# Dynamic Game Analysis of Enterprise Green Technology Innovation Ecosystem under Double Environmental Regulation

**DOI:** 10.3390/ijerph191711047

**Published:** 2022-09-03

**Authors:** Yan Li, Yi Shi

**Affiliations:** School of Economics and Management, Taiyuan University of Technology, Taiyuan 030024, China

**Keywords:** formal environmental regulation, informal environmental regulation, green technological innovation, green finance, evolutionary game

## Abstract

In the context of China’s “double carbon” target, an urgent problem that remains to be solved is how to drive the construction of an enterprise green innovation ecosystem through effective environmental regulations to alleviate the pressure of energy saving and emission reduction. Based on this, we constructed a tripartite evolutionary game model of enterprises, governments and financial institutions, and used the evolutionary game theory and MATLAB simulation to analyze the evolutionary process of the interaction of the subjects of the green technology innovation of enterprises under the dual environmental regulation. The research finds that: (1) Both formal and informal environmental regulations can promote green technology innovation in enterprises, provided that the enforcement is controlled within an appropriate range; (2) Informal environmental regulations are a weaker driver of green technology innovation in firms than formal environmental regulations; (3) Six types of environmental regulation strategies, namely, the “penalty enterprises mechanism“, “financial support mechanism“, “public supervision mechanism”, “punishes financial institutions mechanism”, “financial subsidy mechanism” and “carbon tax mechanism“, have a decreasing effect on promoting the development of the green technology innovation ecosystem of enterprises; (4) Combining the implementation of a middle-intensity subsidy mechanism, a high-intensity penalty mechanism, a low-intensity public supervision mechanism and a middle-intensity carbon tax mechanism is the optimal strategy combination to encourage collaborative green technology innovation between companies and financial institutions.

## 1. Introduction

The 2021 Emissions Gap Report, published by the United Nations Environment Programme (UNEP), states that updated national climate commitments combined with other climate change mitigation measures could put the world on a “2.7 °C global warming trajectory by the end of the century,” which is well above the Paris Agreement’s temperature control target and would trigger catastrophic climate change. To keep global warming below 1.5 °C by the end of the century, meeting the aspirational goals of the Paris Agreement, the world needs to halve its annual greenhouse gas emissions over the next eight years. The Chinese government attaches great importance to the dilemma of quality economic growth and carbon emission reduction, and actively seeks guidelines and policies to promote quality development and ecological protection in a synergistic manner. To achieve the daunting task of high-quality economic growth and carbon emission reduction, powerful measures have been formulated to “strive to peak carbon dioxide emissions by 2030 and strive to achieve carbon neutrality by 2060” [1]. The International Energy Agency (IEA) pointed out that green technology can theoretically contribute to more than 60% of the target carbon reduction, and is expected to become a leading factor in achieving carbon reduction and mitigating climate change [2].

Along with the huge pressure on China’s economic structural reform and development, it is urgent to provide a strong impetus for economic and social development through green technology innovation to achieve the “double carbon” goal. Green technology innovation is different from traditional technology innovation; green technology is a new type of modern technology coordinated with the ecological environment system. Its special attributes and values determine that it can fundamentally solve the contradiction between ecological environmental and economic development, and that it is an important support for the construction of an ecological civilization and high-quality economic development [3].

However, green technology innovation not only has the characteristics of traditional technology innovation, but also the particularity of a not-obvious direct economic effect and long investment cycle. Additionally, differing from traditional technology innovation, the future environmental and ecological benefits generated by green technology innovation often require the sacrifice of immediate economic benefits, which is contrary to the goal of maximizing economic profits and prone to “environmental externality” problems [4]. Additionally, it also has the problem of “innovation externality”, so green technology innovation has the characteristics of “double externality” [5]. The “double externality” often leads to “double market failure”, making green technology innovation an investment under the social optimal scale in the long run [6]. Additionally, constraints such as high innovation risks and financing constraints have significantly inhibited the enthusiasm of new energy enterprises to carry out green technology innovation. The government can solve the problem of “double externalities” and market failure of green technology innovation by adopting effective environmental regulation policies [7]; financial institutions can mitigate the financing constraints of enterprises by providing them with financing services and help to carry out green technology innovation.

Therefore, the synergy of green innovation between financial institutions and enterprises under government environmental regulation is an important guarantee to promote enterprises’ green technology innovation. However, it also faces some problems, such as the moral hazard of some companies defrauding green financial investment [8]; the lack of corporate information disclosure, which leads to the insufficient investment willingness of green financial institutions [9]; the government mismatch problem between policy and market, and the policy efficiency not being high. It is an urgent task for the Chinese government to create a positive and healthy green financial atmosphere, promote green technology innovation and build an effective “green innovation ecosystem”. However, most of the current articles still focus on one of the chains, such as the study of the impact of environmental regulation on green technology innovation [10] and the study of the mechanism of green finance on green innovation, but few articles focus on green innovation ecosystems [11].

In addition, energy resources, an important safeguard in the process of industrialization in China, have played a key role in supporting high-quality economic development. However, China consumes about 20% more coal per unit of thermal power generated than other countries, and the per capita oil reserves and per capita natural gas reserves are also far below the world average. Enterprises being the main body of innovation, the green technology innovation of new energy enterprises is the key to solve such problems. Only by improving the green technology innovation ability of new energy enterprises and changing the economic development mode can we solve the energy bottleneck problem in the process of China’s economic development and ensure the sustainable supply of energy. However, China’s new energy enterprises as a whole have a low level of technological innovation, large disparities within the industry, uneven development levels between regions and other problems, which seriously hinder their smooth and healthy development. Additionally, it is difficult to maintain a sustainable advantage in innovation by virtue of their own advantages and survive in the competitive market environment. Therefore, collaborative innovation with financial institutions and the establishment of enterprise green innovation ecosystems have become the superior choices for new energy enterprises to enhance their green technology innovation capability, development and growth.

This study takes green technology innovation theory, environmental regulation theory, financing constraint theory, green finance theory and evolutionary game theory as the theoretical basis, and uses the evolutionary game method and numerical simulation method to build a tripartite relationship between enterprises, government and financial institutions. The evolutionary game model is used to analyze the optimal strategic path choices of all parties in the “green innovation ecosystem” under dual environmental regulation.

This paper answers the following important questions:Can dual environmental regulation promote collaborative green technology innovation between enterprises and financial institutions, thus addressing the problem of low degrees of coordination between the two due to information asymmetry?Do environmental regulation strategies under different implementation efforts have heterogeneous effects on the choice of strategies among parties in the enterprise green innovation ecosystem?In the enterprise green innovation ecosystem, what choices should the government make to achieve the optimal strategy and optimal execution intensity to promote the active participation of enterprises and financial institutions in the green innovation, so as to achieve the goal of building an effective green innovation ecosystem?

This paper is divided into the following contents: the second part is a literature review, which reviews previous relevant studies; the third part is the influence mechanism analysis, the construction of the evolutionary game model and the analysis of the tripartite evolution stability in the enterprise green innovation system; the fourth part is an evolutionary simulation analysis, which analyzes the heterogeneous effects of different choices of environmental regulation strategies and different implementation efforts on the evolution of the system; the last part is a conclusion and enlightenment, which summarizes the research results and puts forward relevant policy recommendations based on the research findings.

## 2. Literature Review

### 2.1. Green Finance and Green Technology Innovation

As an investment activity, green technology innovation activities are inseparable from the support of financing. Based on the priority-order financing theory and the pecking-order theory, enterprises prefer the endogenous financing method with low cost and high accessibility when conducting financing [12]. However, because the enterprise green technology innovation activity has the characteristics of high investment of innovation funds, indivisibility of innovation process, long and irregular innovation cycle, uncertainty of innovation results and high risk and information asymmetry of each innovation subject, the internal funds generated by the enterprise alone are far from satisfying for the enterprise green technology innovation funds investment demand. Therefore, exogenous financing is an important source of enterprise technology innovation funds [13]. The common problems of insufficient collateral value and information asymmetry will block the external financing channels of enterprises to a certain extent, which will reduce the level of R&D investment and thus make the technological innovation of enterprises more restricted [14]. However, the technological innovation activities of new energy enterprises require more capital and are more dependent on external financing. Based on the capital structure control theory, the imperfection of the capital market makes the external financing cost of new energy enterprises much higher than the internal financing cost of enterprises, which faces a stronger financing constraint [15]. Existing studies suggest that green finance can help solve the financing difficulties of new energy projects with long payback periods and unpredictable risk factors by allocating a financial supply that matches the capital needs of green technology innovation [16,17]. It is crucial to develop green finance to improve green productivity, green technology innovation and low carbon emissions by directing capital flows to green technology innovation [11,18,19].

Therefore, a current research trend is to study the impact of green finance on green technology innovation, and some scholars have recognized that green finance and green technology innovation are not two independent systems, so they have combined the two and explored the relationship between the roles of green finance and green technology innovation.

Most scholars believe that green finance plays a positive role for corporate green technology innovation. By penetrating into various actors of the innovation system, it enhances the risk resistance of enterprises [7], provides financial support and platform support for green technology innovation [20,21] and thus effectively enhances the green innovation capability of enterprises [22]. In terms of mechanism of action, green finance promotes enterprise green technological innovation by increasing the proportion of long-term borrowing and improving corporate debt structure; in terms of heterogeneity of corporate characteristics, the promotion effect of green finance is more significant for non-polluting, large-scale and state-owned enterprises [23]; at the level of financial environment heterogeneity, the promotion effect of this policy is more significant for enterprises in regions with weaker bank competition [23]; in terms of regional heterogeneity, the promotion effect of green finance on corporate green technological innovation is more significant in provinces with rich resource endowment, high levels of economic development and high degrees of marketization [24,25].

In addition, compared with other industries, the financing of the new energy industry is more difficult, and it is difficult for general financial instruments to play a role. It is necessary to give financial support and policy inclination to promote the new energy industry to move towards the mid-to-high end. The role of green finance is highlighted, but considering that the development of the green finance system in China is still immature and faces many problems that need to be solved urgently, it has become an obstacle to the enterprises’ green technology innovation. Additionally, the research on green finance and enterprise green technology innovation in the related literature is still scarce and remains at the empirical research stage. This paper adopts the evolutionary simulation method to overcome the limitations of the existing research methods, introduces financial institutions into the evolutionary game model and uses MATLAB for the numerical simulation analysis, which intuitively reflects the evolutionary drivers and trends of green technology innovation in new energy enterprises under the different strategy changes of financial institutions.

### 2.2. Collaborative Green Technology Innovation between Financial Institutions and Enterprises under Environmental Regulation

The synergy between financial institutions and enterprises’ green technology innovation under environmental regulation is an important guarantee to promote enterprises’ green technology innovation [26], which has now attracted extensive attention from domestic and international research scholars.

The combination of green finance and environmental protection regulations can effectively alleviate the financial pressure brought by the technological upgrading of enterprises, improve the level of their innovative technology and achieve a “win-win” between economic development and environmental protection [27]. Government leadership is still the basis for the development of corporate green innovation ecosystems [28]. The government can not only force enterprises to innovate green technology through environmental regulation strategies [29], but also reduce the cost of green innovation through tax breaks and fiscal transfer payments. However, the challenges faced by enterprises in terms of competitive external environments [30,31], high innovation risks, long investment return cycles [32] and high financing costs [33] have led to the enterprises’ fearfulness in green technology innovation strategies. The information asymmetry further exacerbates the mutual suspicion between financial institutions and enterprises, resulting in a low degree of synergy between financial institutions and enterprises in green innovation. In addition, China has long been accompanied by contradictions such as the lack of synchronization between environmental performance and economic performance, the mismatch between innovation strategy and resource capacity and the overall low performance of green innovation [34]. One of the important reasons is that the implementation of the green innovation paradigm with the synergy of multiple subjects is not in place, the green innovation ecosystem is not well constructed and the interests of relevant subjects are difficult to coordinate [35]. In this context, it is of great practical significance to study how to improve the synergy between financial institutions and enterprises in green technology innovation through environmental regulation, so as to implement the green technology innovation paradigm of multi-subject coordination and achieve the purpose of building an effective green innovation ecosystem.

However, few studies have investigated the coordination mechanism of the three, and most of the existing studies are static. In fact, the green technology innovation of new energy enterprises is a complex engineering system, which is jointly promoted by the government, enterprises and financial institutions. Government regulation is the foundation for the development of enterprise green technology innovation, enterprises are the main body of green transformation and financial institutions provide comprehensive financial services for enterprise green technology innovation. Changes in the strategy choice of any of the participants in enterprise green technology innovation will cause corresponding changes in the response strategies of stakeholders, and such changes are not only the continuous improvement of information structure, but also the adjustment of rational level of multiple participants. Therefore, this paper clearly describes the evolutionary game of interests among the three, portrays the decision-making process of the participants in enterprises’ green technology innovation, reveals the dynamic evolutionary law of the strategies of the three players as well as analyzes and predicts the behavioral decision-making of the participants from the perspective of limited rationality.

### 2.3. Enterprise Green Innovation Ecosystem

Enterprise innovation ecosystem is a very popular topic of academic research, and scholars are currently discussing corporate innovation ecosystem in terms of connotation, operation and evolution. Regarding the connotation of the corporate innovation ecosystem, Moore first proposed the concept of ecosystem in 1993 [36], and then a large number of scholars have successively identified the related concepts, among which innovation ecosystem has attracted great attention from academia [37]. First proposed by Ander, the concept of innovation ecosystem consists of core firms, upstream and downstream firms in the supply chain and users, which together form a closely coordinated and complementary network [38] and can provide greater convenience for core enterprises in the area of resources [39]. However, when the motives of core and upstream and downstream firms are not aligned, the instability of the corporate innovation ecosystem will increase and the corresponding risk level will be higher [40]. Later, it was further proposed that innovation ecosystem is the process of building interdependent relationships among firms to achieve value creation and capture [41]. Enterprises can improve their core competencies by developing and integrating multiple complementary platforms [42] and continuously gain competitive advantage by coordinating the system’s internal and external resources [43].

Corporate green innovation ecosystem means the combination of corporate green innovation and innovation ecology. Since 2005, scholars have gradually integrated the concept of “green” into innovation ecosystems and studied it from micro, macro and systemic perspectives. However, at present, the research is focused on a single chain in the green innovation ecosystem, for example, only on the factors influencing green innovation in enterprises, and the research points out that green innovation activities aim to achieve environmental friendliness and sustainable development [44]. The green innovation system goes beyond the current technological innovation model and includes the effectiveness of green industries and green economy [45], with positive and negative externalities. Both the traditional and dynamic capabilities of a firm can significantly enhance its green innovation performance [46]. When companies conduct green innovation, they face challenges such as high complexity, high investment, high risk and long cycle time, so they need to leverage other companies and investors. Furthermore, some scholars have studied the relationship between core enterprises and upstream and downstream enterprises in the supply chain and found that simultaneous green innovation R&D by upstream and downstream enterprises is a key way to achieve value creation [47]. However, upstream and downstream enterprises rarely conduct green innovation R&D simultaneously, with upstream enterprises focusing on green innovation in green product design and downstream enterprises focusing on R&D activities related to green eco-design [48].

In summary, both innovation ecosystems and green innovation have received widespread academic attention, but the existing studies generally focus on innovation ecosystems in the traditional sense and on the impact factors of green innovation. The characteristics of green technology innovation lead to the uniqueness of the green innovation ecosystem, and the construction of effective green innovation ecosystems is important for high-quality economic development and sustainable development; however, only very few scholars conduct research on green innovation ecosystems.

Moreover, there is still a lack of research on the stability of the green innovation ecosystem and collaborative innovation among multiple agents. Government and financial institutions are important innovation support layers in the green innovation ecosystem of enterprises, but few studies have included the government and financial institutions as important game subjects into the green innovation ecosystem game model for systematic research. At the same time, the government participation approach tends to only consider government financial subsidies and does not further refine them. Therefore, this paper takes the green innovation ecosystem as the research object, and takes the government, financial institutions and new energy enterprises as the main research subjects. From the perspective of dual environmental regulation, we study how to build an effective and healthy long-term stable development of an enterprise green innovation ecosystem.

## 3. Influence Mechanism Analysis, Evolutionary Game Model Construction and Evolutionary Stability Analysis

### 3.1. Analysis of the Impact Mechanism of Dual Environmental Regulation and Green Technology Innovation

Environmental regulation refers to social regulation aimed at reducing pollution and protecting the environment. It can be divided into formal environmental regulation and informal environmental regulation according to the different implementing entities. Among them, formal environmental regulation refers to relying on a series of government administrative orders to manage the environment and to achieve the purpose of environmental governance, that is, formal environmental regulation under the leadership of the government. Informal environmental regulation means that when the formal environmental regulation implemented by the government cannot meet the public’s demand for environmental pollution control, then the informal environmental regulation will serve as an important supplement to the formal environmental regulation, so that the public can achieve higher environmental benefits. This takes place through mass petitions, media exposure and other means to exert pressure on the local government and polluting enterprises, so as to achieve the purpose of improving the living environment and improving the quality of life.

The impact mechanism of dual environmental regulation on green technology innovation is shown in Figure 1.

#### 3.1.1. Mechanisms of the Role of Formal Environmental Regulation on Green Technology Innovation

The mechanisms of formal environmental regulation for green technology innovation are mainly manifested in two aspects. First, when the cost of technological innovation is high for the enterprises, the increase in the intensity of environmental regulation will prompt enterprises to reduce pollution emissions. However, as the cost of environmental regulation rises, the profitability of enterprises decreases, at which time they may choose to misappropriate R&D investment for pollution control, resulting in insufficient financial and material resources for enterprises to carry out technological innovation as well as a decrease in technological innovation efficiency and innovation capacity. Second, when the intensity of environmental regulation is appropriate, enterprises generally innovate production technologies by increasing investment in R&D to improve productivity, although this may lead to an increase in pollution emissions. However, since the improvement of production technology will also significantly increase the profit margin of the enterprise, it can compensate the cost of pollution control.

#### 3.1.2. The Mechanism of Informal Environmental Regulation on Green Technology Innovation

In the context of sustainable development policies, the public’s requirements for environmental quality are constantly improving, and they are paying more attention to environmental pollution. When a pollution incident occurs, the public will exert pressure on local polluting enterprises through a series of means of negotiation, media exposure, petition and appeal, forcing the enterprises to reduce their pollution emissions. At the same time, when the intensity of informal environmental regulation is strong enough, polluting enterprises will increase capital investment to introduce or develop green technologies for the sake of maintaining their own image and social reputation, that is, from “end pollution control” to “source pollution control”. Therefore, informal environmental regulation can also help stimulate enterprises to carry out technological innovation.

### 3.2. Evolutionary Game Theory

Due to the complexity of the real social and economic environment and decision-making issues, it is difficult for the traditional game theory based on the assumption of complete rationality to draw reliable conclusions. As an important analytical tool for bounded rational games, the evolutionary game theory has become one of the most important research fields of modern game theory. The evolutionary game model has the following characteristics: firstly, it takes the participating group as the object of study, analyzes the dynamic evolutionary process and explains why and how the group reaches this state; secondly, the evolution of the group has both selection and mutation processes; thirdly, the behavior selected by the group has a certain inertia.

The evolutionary game theory is based on bounded rationality, and is founded on the basis of the biological evolution theory, which believes that individuals in a group can achieve a stable dynamic equilibrium through imitation, learning, mutation and other processes, and form an evolutionary stable strategy [49]. Evolutionary stable strategy (ESS) and replication dynamics are two important concepts in the evolutionary game theory. The definition of evolutionary stable strategy is: if and only if A is also a Nash equilibrium, E(B, B) is the payoff of both game parties when the strategy is (B, B), then for any arbitrary B has [E(A, A) ≥ E(A, B)]. If a few game parties deviate from A after reaching the steady state of A adopted by all game parties, the optimal response dynamics will cause the game parties’ strategies to quickly return to the state of A adopted by all game parties, so the steady state of A adopted by all game parties is robust and A is an evolutionary stable strategy.

The replication dynamic equation can be represented by the following dynamic differential equation:$$F\left(x_{i}\right)=\frac{dx_{i}}{dt}=x_{i}\left[E_{i}-\bar{E}\right]$$

In the above equation, *x_i_* denotes the proportion of the game population adopting strategy *i*; *E_i_* is the expected value of adopting strategy *i*, which is the average strategy expectation. Let *F*(*x_i_*) = 0 be able to obtain the stability point of the replicated dynamic equation. If the players learn slowly or make mistakes, the original stability will be slightly disturbed and deviated, and the replication dynamics will bring it back to the original stability level. Based on the above theory and focusing on the research topic of this paper, this paper explores the behavioral strategy choices and systematic evolution paths of new energy companies and financial institutions participating in green technology innovation under the perspective of heterogeneous environmental regulation by constructing an evolutionary game model.

### 3.3. Evolutionary Game Model Construction

**Hypothesis** **1.**
*In a “natural” state without considering other influencing factors, there exists an enterprise green innovation ecosystem consisting of new energy enterprises, the government and financial institutions, in which all three participants are limited rational individuals with learning abilities and have their own behavioral options and rights. As shown in Figure 2, in this system, the strategy of new energy enterprises is to carry out green technology innovation or to not carry out green technology innovation, the government’s strategy is active regulation and negative regulation and the strategy of financial institutions is to provide green investment to enterprise green technology innovation or to not provide green investment. The probability of a new energy enterprise to carry out green technology innovation is x, the probability of active government regulation is y, the probability of a financial institution to provide green investment to an enterprise is z, and x, y, z ∈ [0, 1].*


**Hypothesis** **2.**
*The basic income of new energy enterprises when they choose traditional technology is R. When new energy enterprises choose green technology innovation, the increased revenue is R_1_, and the increased benefit of new energy enterprises when the government chooses to carry out environmental regulation is R_2_. When financial institutions choose to invest in the green technology of enterprises, the increased revenue of the new energy enterprises during innovation is R_3_.*


**Hypothesis** **3.**
*In the green innovation ecosystem, government regulation includes dual environmental regulation of formal environmental regulation and informal environmental regulation, the specific assumptions of which are shown in Figure 3. Formal environmental regulation includes three strategies: subsidy mechanism, punishment mechanism and environmental taxation mechanism.*


(1)The subsidy mechanism (positive incentive strategy) consists of two strategies: financial support for enterprises and financial subsidies for financial institutions, hereinafter referred to as “financial support mechanism” and “financial subsidy mechanism”. Financial support is allocated prior to the implementation of green technology innovation activities by enterprises and can subsidize the cost of green technology innovation by enterprises;(2)The punishment mechanism (reverse incentive strategy) includes punishment for the “rent-seeking” behavior of enterprises that receive financial support from the government but do not carry out green technology innovation and punishment for the financial institutions for accepting financial subsidies provided by the government but not fulfilling their obligations to provide financing to enterprises, hereinafter referred to as the “penalty enterprises mechanism” and the “penalty financial institution mechanism”. The government will not only punish such “subsidy fraud” behaviors, but also recover the subsidies;(3)The carbon tax mechanism (reverse incentive strategy) can motivate enterprises to reduce energy consumption, develop alternative energy sources and force them to make green transformations and upgrades. The government will impose a carbon tax on companies that use traditional technologies and exempt them from the tax if they implement green technology innovations.

The execution intensity factors of providing financial support to enterprises, providing financial subsidies to financial institutions, imposing penalties on enterprises, imposing penalties on financial institutions and imposing carbon taxes on enterprises are *α, ε, γ, m* and *n*, and the corresponding costs are *αS, εH, γP, mF* and *nT*.

**Hypothesis** **4.**
*Informal environmental regulation refers to the supervision of enterprises from the public level and the promotion of enterprises to carry out green technology innovation spontaneously, which is hereafter referred to as the “public supervision mechanism”. The government will promote the formation of informal environmental regulation forces by introducing policies such as “green publicity to the public” and “rewarding the public for reporting behavior”. The execution intensity factor of informal environmental regulation is λ, corresponding to the cost consumed which is λI.*


**Hypothesis** **5.**
*Financial institutions mainly adopt two strategies for new energy enterprises. One is to provide financial support for new energy enterprises’ green technology innovation activities and make investments before the enterprises carry out green technology innovation behaviors, which can alleviate the financing constraints of new energy enterprises, and the intensity factor of the financial institutions’ green financial investment in new energy is δ. The upper limit of investment amount is J. Secondly, financial institutions will supervise whether new energy enterprises really carry out green technology innovation if they choose to make green investments, and the supervision cost is C_y_. However, when the government chooses to actively regulate, it discloses whether the enterprise has engaged in green technology innovation, and the financial institutions will not have to pay the supervision cost. Whether it is government monitoring or financial institutions monitoring, once it is found that enterprises accept green investment but do not develop green technology innovation, they will be publicly denounced, and not only will the subsequent financing of the enterprises be seriously impacted, but the enterprises will also bear the loss of reputation decline for T_2_.*


**Hypothesis** **6.**
*The innovation cost consumed by the green technology innovation of new energy enterprises is C_m_*
*, and the input cost required by the government in choosing to make environmental regulation is C_n_. One can perceive the gain P_n_ brought by new energy enterprises’ green technology innovation, or the loss S_n_ brought by new energy enterprises not making green technology innovation. The perceived gain to the government from financial institutions making green investments is P_g_, and the perceived loss to the government from financial institutions not making green investments is S_g_. The original basic income of financial institutions is P_y_, and when they choose to make green investment in enterprises, they can perceive the increase in perceived gain ΔP_y_ from green technology innovation by new energy enterprises or they can perceive the perceived loss S_y_ caused by new energy enterprises not conducting green technology innovation.*


The influence mechanisms of the parties within the corporate green innovation ecosystem are shown in Figure 4. The parameters set in hypotheses 1 to 6 are shown in Table 1.

The payment matrix of the evolutionary game between the new energy enterprises, government and financial institutions is constructed based on the above assumptions, as shown in Table 2.

Let *E*_11_ and *E*_12_ represent, respectively, the expected benefits of “Green technology innovation” and “No green technology innovation” for new energy enterprises. According to Table 2, the payoffs of the new energy enterprises with the two different behavior strategies are as follows:(1)$$E_{11}=z\left(\delta J+R_{3}\right)+y\left(aS+\lambda I+R_{2}\right)+\left(R+R_{1}-C_{m}\right)$$
(2)$$E_{12}=z\left(\delta J-T_{2}\right)+y\left(nT-\gamma P\right)+R$$

The average expected earning of the new energy enterprises is shown as follows:(3)$$\bar{E_{1}}=xE_{11}+\left(1-x\right)E_{12}$$

The replication dynamic equations of the new energy enterprises is shown as follows:(4)$$\begin{array}{ll} F(x) & =\frac{\mathrm{dx}}{dt}=x\left(E_{11}-\bar{E_{1}}\right)\\ & =x\left(1-x\right)\left[z\left(T_{2}+R_{3}\right)+y\left(aS+\lambda I+nT+\gamma P+R_{2}\right)+\left(R_{1}-C_{m}\right)\right] \end{array}$$

Let *E*_21_ and *E*_22_ represent, respectively, the expected earnings of “Active regulation” and “Negative regulation” for the government. According to Table 2, the payoffs of the government with the two different behavior strategies are as follows:(5)$$E_{21}=z\left(-\varepsilon H+P_{g}-mF+S_{g}\right)+x\left(-\alpha S-\lambda I-nT-\gamma P+P_{n}+S_{n}\right)+\left(nT+\gamma P+mF-S_{n}-S_{g}-C_{n}\right)$$
(6)$$E_{22}=0$$

The average earning of the government is shown as follows:(7)$$\bar{E_{2}}=yE_{21}+(1-y)E_{22}$$

The replication dynamic equations of the government are shown as follows:(8)$$\begin{array}{ll} F(y) & =\frac{dy}{dt}=y\left(E_{21}-\bar{E_{2}}\right)\\ & =y\left(1-y\right)\left[z\left(-\varepsilon H+P_{g}-mF+S_{g}\right)+x\left(-\alpha S-\lambda I-nT-\gamma P+P_{n}+S_{n}\right)+\left(nT+\gamma P+mF-S_{n}-S_{g}-C_{n}\right)\right] \end{array}$$

Let *E*_31_ and *E*_32_ represent, respectively, the expected earnings of “Green investment” and “No green investment” for the financial institutions. According to Table 2, the payoffs of the financial institutions with the two different behavior strategies are as follows:(9)$$E_{31}=x\left(\Delta P_{y}+S_{y}\right)+y\left(\varepsilon H+C_{y}\right)+\left(\begin{array}{c} P_{y}-\delta J-C_{y}-S_{y} \end{array}\right)$$
(10)$$E_{32}=-ymF+P_{y}$$

The average earning of the financial institution is shown as follows:(11)$$\bar{E_{3}}=zE_{31}+\left(1-Z\right)E_{32}$$

The replication dynamic equations of the financial institution are shown as follows:(12)$$\begin{array}{ll} F\left(z\right) & =\frac{dz}{dt}=z\left(E_{31}-\bar{E_{3}}\right)\\ & =z\left(1-z\right)\left[x\left(\Delta P_{y}+S_{y}\right)+y\left(\varepsilon H+C_{y}-mF\right)+\left(-\delta J-S_{y}-C_{y}\right)\right] \end{array}$$

### 3.4. Analysis of Trilateral Evolutionary Stabilization Strategies

To find the equilibrium point of the evolutionary stabilization strategies, let *F*(*x*) = 0, *F*(*y*) = 0 and *F*(*z*) = 0. The evolutionary stabilization equilibrium points of the new energy enterprises, government and financial institution can be obtained, respectively: *E*_1_ = (0, 0, 0), *E*_2_ = (0, 1, 0), *E*_3_ = (0, 0, 1), *E*_4_= (1, 0, 0), *E*_5_ = (0, 1, 1), *E*_6_ = (1, 1, 0), *E*_7_ = (1, 0, 1), *E*_8_ = (1, 1, 1), *E*_9_ = (x^*^, y^*^, z^*^), and $x^{*}=\frac{-y\left(\varepsilon H+C_{y}-mF\right)+\left(\delta J+S_{y}+C_{y}\right)}{\Delta Py+Sy}$, $y^{*}=\frac{-z\left(T_{2}+R_{3}\right)-\left(R_{1}-C_{m}\right)}{\alpha S+\lambda I+nT+\gamma P+R_{2}}$, $z^{*}=\frac{x\left(\alpha S+\lambda I+nT+\gamma P-P_{n}-S_{n}\right)-\left(nT+\gamma P+mF-S_{n}-S_{g}-C_{n}\right)}{-\varepsilon H+P_{g}-mF+S_{g}}$.

#### 3.4.1. Evolutionary Stability Analysis for Enterprises

Let *F*(*x*) = 0, get $y^{*}=\frac{-z\left(T_{2}+R_{3}\right)-\left(R_{1}-C_{m}\right)}{\alpha S+\lambda I+nT+\gamma P+R_{2}}$.

Situation 1: If $y=\frac{-z\left(T_{2}+R_{3}\right)-\left(R_{1}-C_{m}\right)}{\alpha S+\lambda I+nT+\gamma P+R_{2}}$, then *F*(*x*) = 0; x takes any value in the interval is a steady state and the probability of the enterprises’ strategy choice x does not change over time.

Situation 2: If $y\neq \frac{-z\left(T_{2}+R_{3}\right)-\left(R_{1}-C_{m}\right)}{\alpha S+\lambda I+nT+\gamma P+R_{2}}$, then *F(x*) = 0, x = 0 and x = 1 are two strategies for x.

The derivation of *F(x)* is obtained:(13)$$\frac{\partial F\left(x\right)}{\partial x}=\left(1-2x\right)\left[z\left(T_{2}+R_{3}\right)+y\left(\alpha S+\lambda I+nT+\gamma P+R_{2}\right)+\left(R_{1}-C_{m}\right)\right]$$

When $y<\frac{-z\left(T_{2}+R_{3}\right)-\left(R_{1}-C_{m}\right)}{\alpha S+\lambda I+nT+\gamma P+R_{2}}$, there are $\frac{\partial F\left(x\right)}{\partial x}{|}_{x=0}<0$ and $\frac{\partial F\left(x\right)}{\partial x}{|}_{x=1}>0$. So x = 0 is the evolutionary equilibrium point, that is, when the probability of environmental regulation by the government is less than $\frac{-z\left(T_{2}+R_{3}\right)-\left(R_{1}-C_{m}\right)}{\alpha S+\lambda I+nT+\gamma P+R_{2}}$, new energy enterprises choose not to carry out green technology innovation. In this case, the initial state of the game is located in the space V_11_.

When $y>\frac{-z\left(T_{2}+R_{3}\right)-\left(R_{1}-C_{m}\right)}{\alpha S+\lambda I+nT+\gamma P+R_{2}}$, there are $\frac{\partial F(x)}{\partial x}{|}_{x=1}<0$ and $\frac{\partial F(x)}{\partial x}{|}_{x=0}>0$. So x = 1 is the evolutionary equilibrium point, that is, when the probability of environmental regulation by the government is greater than $\frac{-z\left(T_{2}+R_{3}\right)-\left(R_{1}-C_{m}\right)}{\alpha S+\lambda I+nT+\gamma P+R_{2}}$, the new energy enterprises choose to carry out green technology innovation. In this case, the initial state of the game is located in the space V_12_.

Based on the above analysis, the replicated dynamic phase diagram of the new energy enterprise is obtained as shown in Figure 5.

#### 3.4.2. Evolutionary Stability Analysis for Government

Let *F*(*y*) = 0, get $x^{*}=\frac{-z\left(-\varepsilon H+P_{g}-mF+S_{g}\right)-\left(nT+\gamma P+mF-S_{n}-S_{g}-C_{n}\right)}{\left(-\alpha S-\lambda I-nT-\gamma P+P_{n}+S_{n}\right)}$.

Situation 1: If $x=\frac{-z\left(-\varepsilon H+P_{g}-mF+S_{g}\right)-\left(nT+\gamma P+mF-S_{n}-S_{g}-C_{n}\right)}{\left(-\alpha S-\lambda I-nT-\gamma P+P_{n}+S_{n}\right)}$, then *F*(*y*) = 0; y takes any value in the interval is a steady state and the probability of the governments’ strategy choice y does not change over time.

Situation 2: If $x\neq \frac{-z\left(-\varepsilon H+P_{g}-mF+S_{g}\right)-\left(nT+\gamma P+mF-S_{n}-S_{g}-C_{n}\right)}{\left(-\alpha S-\lambda I-nT-\gamma P+P_{n}+S_{n}\right)}$, then *F*(*y*) = 0, y = 0 and y = 1 are two strategies for y.

The derivation of *F*(*y*) is obtained:(14)$$\frac{\partial F\left(y\right)}{\partial y}=\left(1-2y\right)\left[z\left(-\varepsilon H+P_{g}-mF+S_{g}\right)+x\left(-\alpha S-\lambda I-nT-\gamma P+P_{n}+S_{n}\right)+\left(nT+\gamma P+mF-S_{n}-S_{g}-C_{n}\right)\right]$$

When $x<\frac{-z\left(-\varepsilon H+P_{g}-mF+S_{g}\right)-\left(nT+\gamma P+mF-S_{n}-S_{g}-C_{n}\right)}{\left(-\alpha S-\lambda I-nT-\gamma P+P_{n}+S_{n}\right)}$, there are $\frac{\partial F\left(y\right)}{\partial y}{|}_{y=0}<0$ and $\frac{\partial F\left(y\right)}{\partial y}{|}_{y=1}>0$. So y = 0 is the evolutionary equilibrium point, that is, when the probability of green technology innovation by new energy enterprise is less than $\frac{-z\left(-\varepsilon H+P_{g}-mF+S_{g}\right)-\left(nT+\gamma P+mF-S_{n}-S_{g}-C_{n}\right)}{\left(-\alpha S-\lambda I-nT-\gamma P+P_{n}+S_{n}\right)}$, the government chooses negative environmental regulation. In this case, the initial state of the game is located in the space V_21_.

When $x>\frac{-z\left(-\varepsilon H+P_{g}-mF+S_{g}\right)-\left(nT+\gamma P+mF-S_{n}-S_{g}-C_{n}\right)}{\left(-\alpha S-\lambda I-nT-\gamma P+P_{n}+S_{n}\right)}$, there are $\frac{\partial F\left(y\right)}{\partial y}{|}_{y=1}<0$ and $\frac{\partial F\left(y\right)}{\partial y}{|}_{y=0}>0$. So y = 1 is the evolutionary equilibrium point, that is, when the probability of green technology innovation by new energy enterprise is greater than $\frac{-z\left(-\varepsilon H+P_{g}-mF+S_{g}\right)-\left(nT+\gamma P+mF-S_{n}-S_{g}-C_{n}\right)}{\left(-\alpha S-\lambda I-nT-\gamma P+P_{n}+S_{n}\right)}$, the government chooses to activate environmental regulation. In this case, the initial state of the game is located in the space V_22_.

Based on the above analysis, the replicated dynamic phase diagram of the government is obtained as shown in Figure 6.

#### 3.4.3. Evolutionary Stability Analysis for Financial Institutions

Let *F*(*z*) = 0, get $y^{*}=\frac{-x\left(\Delta P_{y}+S_{y}\right)+\left(\delta J+S_{y}+C_{y}\right)}{\varepsilon H+C_{y}-mF}$.

Situation 1: If $y=\frac{-x\left(\Delta P_{y}+S_{y}\right)+\left(\delta J+S_{y}+C_{y}\right)}{\varepsilon H+C_{y}-mF}$, then *F*(*z*) = 0; z takes any value in the interval is a steady state and the probability of the financial institutions’ strategy choice z does not change over time.

Situation 2: If $y\neq \frac{-x\left(\Delta P_{y}+S_{y}\right)+\left(\delta J+S_{y}+C_{y}\right)}{\varepsilon H+C_{y}-mF}$, then *F*(*z*) = 0, z = 0 and z = 1 are two strategies for z.

The derivation of *F*(*z*) is obtained:(15)$$\frac{\partial F\left(z\right)}{\partial z}=\left(1-2z\right)\left[x\left(\Delta P_{y}+S_{y}\right)+y\left(\varepsilon H+C_{y}-mF\right)+\left(-\delta J-S_{y}-C_{y}\right)\right]$$

When $y<\frac{-x\left(\Delta P_{y}+S_{y}\right)+\left(\delta J+S_{y}+C_{y}\right)}{\varepsilon H+C_{y}-mF}$, there are $\frac{\partial F\left(z\right)}{\partial z}{|}_{z=0}<0$ and $\frac{\partial F\left(z\right)}{\partial z}{|}_{z=1}>0$. So z = 0 is the evolutionary equilibrium point, that is, when the probability of environmental regulation by the government is less than $\frac{-x\left(\Delta P_{y}+S_{y}\right)+\left(\delta J+S_{y}+C_{y}\right)}{\varepsilon H+C_{y}-mF}$, the financial institutions choose not to green invest for the new energy enterprise. In this case, the initial state of the game is located in the space V_31_.

When $y>\frac{-x\left(\Delta P_{y}+S_{y}\right)+\left(\delta J+S_{y}+C_{y}\right)}{\varepsilon H+C_{y}-mF}$, there are $\frac{\partial F\left(z\right)}{\partial z}{|}_{z=1}<0$ and $\frac{\partial F\left(z\right)}{\partial z}{|}_{z=0}>0$. So z = 1 is the evolutionary equilibrium point, that is, when the probability of environmental regulation by the government is greater than $\frac{-x\left(\Delta P_{y}+S_{y}\right)+\left(\delta J+S_{y}+C_{y}\right)}{\varepsilon H+C_{y}-mF}$, the financial institutions choose to green invest for the new energy enterprise. In this case, the initial state of the game is located in the space V_32_.

Based on the above analysis, the replicated dynamic phase diagram of the financial institutions is obtained as shown in Figure 7.

### 3.5. Analysis of the Evolutionary Stability of Green Innovation Ecosystems

The replication dynamic equations of the enterprises, government and financial institutions are combined to obtain the replication dynamic system of the enterprise green innovation ecosystem, as shown in Equation (16). In the enterprise green innovation replication dynamic system, the local equilibrium points are not necessarily the evolutionary and stable results of the system. From Lyapunov’s first method, the equilibrium point is asymptotically stable when all the eigenvalues of the Jacobi matrix are satisfied with negative real parts. The Jacobian matrix of the system can be obtained from the replication dynamic system (16) of enterprise green innovation, as shown in Equation (17).
(16)$$\begin{array}{l} \{\begin{array}{l} F\left(x\right)=x\left(1-x\right)\left[z\left(T_{2}+R_{3}\right)+y\left(aS+\lambda I+nT+\gamma P+R_{2}\right)+\left(R_{1}-Cm\right)\right]\\ F\left(y\right)=y\left(1-y\right)\left[z\left(-\varepsilon H+P_{g}-mF+S_{g}\right)+x\left(-\alpha S-\lambda I-nT-\gamma P+P_{n}+S_{n}\right)+\left(nT+\gamma P+mF-S_{n}-S_{g}-C_{n}\right)\right]\\ F\left(z\right)=z\left(1-z\right)\left[x\left(\Delta P_{y}+S_{y}\right)+y\left(\varepsilon H+C_{y}-mF\right)+\left(-\delta J-S_{y}-C_{y}\right)\right] \end{array} \end{array}$$
(17)$$\begin{array}{ll} J & =\left[\begin{array}{ccc} \frac{\partial F\left(x\right)}{\partial x} & \frac{\partial F\left(x\right)}{\partial y} & \frac{\partial F\left(x\right)}{\partial z}\\ \frac{\partial F\left(y\right)}{\partial x} & \frac{\partial F\left(y\right)}{\partial y} & \frac{\partial F\left(y\right)}{\partial z}\\ \frac{\partial F\left(z\right)}{\partial x} & \frac{\partial F\left(z\right)}{\partial y} & \frac{\partial F\left(z\right)}{\partial z} \end{array}\right]\\ & =\left[\begin{array}{ccc} \left(1-2x\right)\left[\begin{array}{l} z\left(T+R_{3}\right)\\ +y\left(\alpha S+\lambda I+nT+\gamma P+R_{2}\right)\\ +\left(R_{1}-Cm\right) \end{array}\right] & x\left(1-x\right)\left(\alpha S+\lambda I+nT+\gamma P+R_{2}\right) & x\left(1-x\right)\left(T+R_{3}\right)\\ y\left(1-y\right)\left(-\alpha S-\lambda I-nT-\gamma P+P_{n}+S_{n}\right) & \left(1-2y\right)\left[\begin{array}{l} z\left(-\varepsilon H+P_{g}-mF+S_{g}\right)\\ +x\left(-\alpha S-\lambda I-nT-\gamma P+P_{n}+S_{n}\right)\\ +\left(nT+\gamma P+mF-S_{n}-S_{g}-C_{n}\right) \end{array}\right] & y\left(1-y\right)\left(-\varepsilon H+Pg-mF+S_{g}\right)\\ z\left(1-z\right)\left(\Delta P_{y}+S_{y}\right) & z\left(1-z\right)\left(\varepsilon H+C_{y}-mF\right) & \left(1-2z\right)\left[\begin{array}{l} x\left(\Delta P_{y}+S_{y}\right)\\ +y\left(\varepsilon H+C_{y}-mF\right)\\ +\left(-\delta J-S_{y}-C_{y}\right) \end{array}\right] \end{array}\right] \end{array}$$

The following takes *E*_1_(0, 0, 0) as an example to analyze its asymptotic stability, and its Jacobian matrix is $J_{1}=\left[\begin{array}{ccc} R_{1}-C_{m} & 0 & 0\\ 0 & nT+\gamma P+mF-S_{n}-S_{g}-C_{n} & 0\\ 0 & 0 & -\delta J-S_{y}-C_{y} \end{array}\right]$.

It is known that the eigenvalues of *E*_1_(0,0,0) are *λ*_1_ = *R*_1_−*C*_m_, *λ*_2_ = *nT* + *γP* + *mF* − *S*_n_ − *S_g_ − C_n_*, *λ*_3_ = −*δJ* − *S_y_ − C_y_*. Similarly, the eigenvalues of the Jacobian matrix can be obtained by bringing the other eight equilibrium points into the Jacobian matrix, as shown in Table 3.

Due to the complexity of the model parameters, this study first assumes that the corresponding parameters satisfy specific conditions when the interests of all parties are satisfied, in order to make the analysis of the stability of corporate green innovation ecosystems simple and without loss of generality. The government, as a finite rational economic man, should invest less in environmental regulation than the sum of the perceived benefits to the government from green technology innovation by firms and green investment by financial institutions in enterprises, setting P_g_ + P_n_ − εH − αS − λI − C_n_ > 0. In the same way, as a bounded rational economic man, when the government supports enterprises to carry out green technology innovation, new energy enterprises must obtain more benefits from green technology innovation than the cost they invest in green technology innovation; otherwise, the enterprises will not choose green technological innovation, setting R_1_ + R_2_ > C_m_. The financial institutions, as finite rational economic agents, should be less penalized than the green subsidies they receive for not providing green investments to enterprises; otherwise, the financial institutions would not risk not providing green investments, setting εH > mF.

The stability of the equilibrium point of the enterprise green innovation replication dynamics system is shown in Table 4.

Combined constraints: P_g_ + P_n_ − εH − αS − λI − C_n_ > 0, R_1_ + R_2_ > C_m_, εH > mF.

**Conclusion I****:** When R_1_ − C_m_ > 0 or R_1_ − C_m_ < 0 and nT + γP+ mF−S_n_ −S_g_ −C_n_ > 0, the enterprise green innovation replication power system’s equilibrium stabilization solution is (1, 1, 1), that is, green technology innovation, active regulation, green investment. At this point, the conditions T_2_ + αS + λI + nT + γP + R_1_ + R_2_ + R_3_ − C_m_ > 0, P_g_ + P_n_ > εH + αS + λI + C_n_ and ΔP_y_ + εH − mF − δJ > 0 are satisfied. That is, for new energy enterprises, the cost invested by enterprises in green technology innovation is less than the increased benefit of green technology innovation by enterprises with government support and investment from financial institutions. If enterprises choose to carry out green technology innovation, then more profits will flow into the enterprises. Therefore, enterprises choose a green technology innovation strategy. For the government, if both firms and financial institutions were actively involved in innovation, there would be substantial perceived benefits to the government that would more than offset the cost of government investment in enterprises and financial institutions. Therefore, the government would choose the active environmental regulation strategy. For the financial institutions, the benefits that the financial institutions can derive from the effects of technological innovation by the firm and green subsidies by the government are greater than the sum of their green investments in enterprises. Therefore, the financial institutions choose the strategy of green investment. 

**Conclusion II****:** When R_1_ − C_m_ < 0 and nT + γP+ mF−S_n_ −S_g_ −C_n_ < 0, the enterprise green innovation replication power system’s equilibrium stabilization solution is (0, 0, 0) and (1, 1, 1), that is, no green technology innovation, negative regulation, no green investment and green technology innovation, active regulation, green investment, respectively. First, when the conditions R_1_ < C_m_, nT + γP+ mF < S_n_+ S_g_ + C_n_ and −δJ − S_y_ − C_y_ < 0 are satisfied, (0, 0, 0) is the equilibrium-stable solution. That is, the cost of investing in green technology innovation by enterprises is too large and far outweighs the benefits generated by their green technology innovation. Not only can it not improve the profitability of the enterprise, but it will also increase the financial burden of the enterprise, which makes it difficult for the enterprise to maintain the green technology innovation strategy and finally choose traditional technology, despite the fact that the government will penalize enterprises or financial institutions that receive subsidies but fail to meet their responsibilities. However, it is difficult to compensate for the losses brought to the government by the non-participation of enterprises and financial institutions in green technology innovation and the costs invested by the government in actively pursuing environmental regulation. Therefore, the government’s behavior evolves into negative regulation. For the financial institutions, when the cost of green investment and supervision is too high, they cannot afford the perceived loss caused by enterprises not innovating green technologies. Therefore, the financial institutions’ behavior evolves into non-green investments. Second, when the conditions T_2_ + αS + λI + nT + γP + R_1_ + R_2_ + R_3_−C_m_ > 0, P_g_ + P_n_ > εH + αS+ λI +C_n_ are satisfied, (1, 1, 1) is the equilibrium-stable solution. This is consistent with Conclusion I, so we do not discuss it in depth again.

## 4. Evolutionary Simulation Analysis

In order to analyze the running trajectories of enterprises’ green technology innovation under different intensities of dual environmental regulation and green financial support, this paper conducts a simulation analysis based on the MALTAB software according to the replicated dynamic equations and constraints. In order to more clearly compare and analyze the heterogeneous effects of different choices of dual environmental regulation strategies on the evolution of the system, the strategies under formal and informal environmental regulations are set to the same upper limit. That is, P = S = T = I = H = F = 2.

On this basis, in order to ensure the robustness of the system evolution results, combined with the interests of new energy enterprises in China, the cost of green technology innovation by enterprises in the short term is higher than the benefit, set R_1_- C_m_ < 0. Other parameters are set as follows: J = 2, R = 10, R_1_ = 6, R_2_ = 4, R_3_ = 3, C_m_ = 8, C_n_ = 4, C_y_ = 3, T_2_ = 1, P_n_ = 6.5, S_n_ = 4.5, P_y_ = 8, ΔP_y_ = 4, S_y_ = 2.5, P_g_ = 4, S_g_ = 2.

The following are the effects when changing the implementation intensity factor of formal and informal environmental regulations on the evolution path.

### 4.1. Evolutionary Paths under Changes in the Initial Willingness of Parties within the Corporate Green Innovation Ecosystem

In the enterprise green innovation ecosystem, the initial willingness of new energy enterprises (x), government (y) and financial institutions (z) varies between 0.1 and 0.9, and the variation interval is set to 1.

In Figure 8, the *x*-axis represents new energy enterprises, the *y*-axis represents the government and the *z*-axis represents the financial institutions. This suggests that as the initial willingness increases, new energy enterprises will eventually stabilize in the “green technology innovation” strategy. Likewise, with the increase in evolution numbers, the 3D simulation diagram objectively shows the apparent evolutionary trend of the strategies of the parties within the enterprise green innovation ecosystem. Finally, the governments will choose to carry out environmental regulation and the financial institutions will choose to provide green investments to enterprises.

### 4.2. The Effect of Changes in Formal Environmental Regulation Enforcement Intensity Factors on the Evolutionary Path

#### 4.2.1. Impact of Different Intensity Subsidy Mechanisms on Evolutionary Paths

(1)Financial support for different implementation intensities

The simulation results of the evolutionary game under the variation of the intensity factor of financial support are shown in Figure 9.

The tripartite evolutionary trajectory of the enterprise innovation ecosystem under different intensities of financial support is shown in Figure 9a. The government provides financial support for enterprises to reduce their green technology innovation input costs. When the implementation intensity factor of financial support is small, the enterprises will not choose to carry out green technology innovation and the financial institutions will not invest in green innovation. When the intensity factor of financial support is larger (above 0.4), new energy enterprises will evolve towards green technology innovation, and the financial institutions will evolve towards investing in enterprises’ green technology innovation, and finally reach stability. That is to say, middle-intensity and high-intensity financial support are able to stimulate enterprises to carry out green technological innovation.

The change of strategy choice of new energy enterprises over time is shown in Figure 9b. When the financial support enforcement intensity factor is small, the enterprises’ willingness to carry out green innovation is enhanced in the short term, but in the long term the enterprises eventually do not choose to carry out green technology innovation; as the financial support enforcement intensity increases, the enterprises evolve toward green technology innovation, and when the enforcement intensity factor reaches 0.4 and above, the enterprises finally stabilize in green technology innovation. Thus, it seems that the financial support can effectively promote enterprises to carry out green technology innovation, and the direct incentive effect is significant. Additionally, the stronger the enforcement, the more obvious the incentive effect on green technology innovation. This is because when enterprises can apply for a large amount of financial support, they can offset most of the green innovation input costs, resulting in a greater “innovation compensation” effect, thus increasing the willingness of enterprises to carry out green technology innovation.

The change in government strategy choice over time is shown in Figure 9c. When the financial support implementation is low, the government evolves toward a positive regulation and the rate of evolution accelerates with the increase in the innovation incentive subsidy coefficient and eventually reaches stability. When the financial support implementation intensity is too high (greater than 0.6), the government chooses a negative regulation strategy due to excessive expenditures. Therefore, the government has to keep the financial support implementation intensity factor at a medium or low level to promote the green technology innovation by enterprises.

The change of financial institutions’ strategy choice over time is shown in Figure 9d. The financial institutions’ strategic choices result in different behavioral choices in response to changes in the implementation of financial support. When the subsidy intensity is low (0.4 and below), the financial institutions do not invest in corporate green innovation. When the subsidy intensity is high (0.6 and above), the financial institutions evolve towards investing in corporate green innovation, and the evolution rate becomes faster with the increase in financial support implementation, and finally reaches a stable state. Therefore, the government’s high-intensity innovation incentive subsidy can well motivate financial institutions to invest in corporate green innovation.

In summary, the medium-intensity financial support is more conducive to motivating firms to engage in green technology innovation. The low intensity of financial support is not enough to produce the “innovation compensation” effect for enterprises, nor can it motivate financial institutions to invest in green innovation of enterprises; the high intensity of financial support will cause excessive financial burden to the government and discourage the government from subsidizing enterprises.

(2)Financial subsidies for different implementation intensities

The simulation results of the evolutionary game under the change of the intensity factor of financial subsidies from the government to the financial institutions are shown in Figure 10.

The three-way evolutionary trajectory of enterprise innovation ecosystem under different intensities of financial subsidies is shown in Figure 10a. The government will provide financial subsidies to financial institutions in order to encourage financial institutions to help enterprises’ green technology innovation, and the intensity factor of government financial subsidies implementation has a significant impact on the decision of the financial institutions. If the government does not provide financial subsidies, the financial institutions do not choose to make green investments in enterprises, and new energy enterprises do not choose to make green technology innovation. When the financial subsidies implementation intensity factor is moderate (0.4–0.6), the enterprises evolve toward green technology innovation and the financial institutions evolve toward investing in the green technology innovation of enterprises, and the speed of stabilization becomes faster with the increase in the financial subsidies intensity until it reaches final stability. When the financial subsidies coefficient is too large (0.8–1.0), the financial institutions evolve toward providing green investments, but the government chooses negative regulation due to the high cost. In other words, medium-intensity financial subsidies are a better incentive for firms to engage in green technological innovation.

The change in strategy choice of new energy enterprises over time is shown in Figure 10b. Initially, the government does not provide financial subsidies to financial institutions or the financial subsidies implementation intensity factor is small (0.2), which indirectly causes enterprises not to engage in green technology innovation. When the financial subsidies reach a certain implementation intensity (0.4–1.0), they can stimulate financial institutions to invest in enterprises; however, when the financial subsidies implementation intensity is too high (0.8–1.0), it affects the government’s enthusiasm to carry out environmental regulation due to excessive government expenditure. Therefore, when the government’s financial subsidies to financial institutions are in the appropriate range (0.4–0.6), it stimulates enterprises to carry out green technological innovation.

The change in government strategy choice over time is shown Figure 10c. When the government has a low (0–0.2) or moderate (0.4–0.6) intensity factor for the implementation of financial subsidies to financial institutions, the government evolves to actively carry out environmental regulation. The smaller the financial subsidies enforcement intensity factor, the faster it evolves and eventually reaches stability. When the financial subsidies enforcement intensity factor is too large (0.8–1.0), the government eventually chooses negative regulation due to excessive expenditures. Therefore, the government’s financial subsidies implementation intensity factor for financial institutions should be kept at a medium level in order to promote green technological innovation by enterprises.

The change of financial institutions’ strategy choices over time is shown in Figure 10d. The strategy choice of financial institutions will generate different behavioral choices as the intensity of government financial subsidies implementation changes. When the financial subsidies implementation intensity factor is low, the financial institutions will not invest in green technology innovation of enterprises; when the financial subsidies implementation intensity factor is high, the direction of the financial institutions to provide green investment to enterprises evolves, and the evolution rate becomes faster with the increase in the financial subsidies implementation intensity factor, and finally reaches a stable state. Therefore, the government’s high-intensity financial subsidies can better motivate financial institutions to provide green investments for new energy enterprises.

#### 4.2.2. Impact of Different Intensity Penalty Mechanisms on Evolutionary Paths

(1)Changes in government enforcement intensities of penalties on enterprises

The simulation results of the evolutionary game under the variation of the intensity factor of government penalties on enterprises are shown in Figure 11.

The tripartite evolutionary trajectory of the enterprise innovation ecosystem under different intensities of government penalties on enterprises is shown in Figure 11a. When the implementation intensity factor of government penalties for enterprises is small, the enterprises will not choose to carry out green technology innovation and the financial institutions will not invest in green innovation. When the intensity factor of government penalties on enterprises is larger (above 0.6), the enterprises will evolve towards green technology innovation, the financial institutions will evolve towards investing in enterprises green technology innovation, and the government will evolve towards environmental regulation and finally reach stability. That is to say, high-intensity government penalties on enterprises are able to stimulate enterprises to carry out green technological innovation.

The change in strategy choice of enterprises over time is shown in Figure 11b. When the enforcement intensity factor of government penalties on enterprises is small (0–0.4), the cost of green technology innovation is high, and the government’s punishment for companies that receive innovation incentive subsidies but do not engage in green technology innovation is small, so the enterprises do not choose to engage in green technology innovation; as the enforcement intensity of government penalties on enterprises increases moderately, the enterprises are in a wait-and-see state at first. Over time, the enterprises perceive the benefits of green technology innovation and evolve towards it. When the enforcement intensity factor reaches 0.6 and above, the new energy enterprises finally stabilize in a green technology innovation strategy. The higher the enforcement intensity of the government penalties on the enterprises, the faster the green technology innovation rate of the new energy enterprises. Thus, it seems that the government penalties on enterprises can effectively motivate enterprises to carry out green technology innovation.

The change in government strategy choice over time is shown in Figure 11c. When the enforcement factor of government penalties on enterprises is small, the enterprises choose traditional technology due to the high cost of green technology innovation, and the government chooses not to carry out environmental regulation strategy due to the high cost of environmental regulation. When the enforcement factor of government penalties on enterprises reaches a suitable range, the government evolves towards actively carrying out environmental regulation, and the speed of evolution accelerates with the increase in the intensity of punishment for enterprises and eventually stabilizes.

The change of the financial institutions’ strategy choice over time is shown in Figure 11d. When the enforcement intensity factor of government penalties on enterprises is small (0–0.4), the financial institutions choose not to provide green investment to enterprises considering the investment risk; when the enforcement intensity factor of government penalties on enterprises is large (0.6–1.0), the financial institutions invest in the green technology innovation of enterprises because the benefit they get is higher than the investment risk. Additionally, the evolution rate becomes faster with the increase in government penalties on enterprises enforcement intensity factor until finally reaching the steady state.

(2)Changes in government enforcement intensity of penalties on financial institutions

The simulation results of the evolutionary game under the variation of the penalty intensity factor of the government to the financial institutions are shown in Figure 12.

The three-way evolutionary trajectory of the enterprise innovation ecosystem under different penalty intensities is shown in Figure 12a. When the penalty enforcement intensity factor is small (0–0.4), the new energy enterprises choose not to carry out green technology innovation, the government chooses not to carry out environmental regulation and the financial institutions choose not to carry out green investment. As the penalty enforcement intensity factor increases (0.6–1.0), the new energy enterprises evolve toward green technology innovation, the government evolves toward active environmental regulation and the financial institutions evolve toward providing green investment, which eventually stabilizes. In other words, the high-intensity penalty intensity factor is more likely to promote green technology innovation.

The change in strategy choice of new energy firms over time is shown in Figure 12b. In the initial period, when the government punishment intensity factor for financial institutions is small (0–0.4), the financial institutions will take the risk not to invest in enterprises green technology innovation, which leads to enterprises not carrying out green technology innovation. When the government punishment to financial institutions reaches a certain enforcement intensity (0.6–1.0), it can reversely stimulate the enthusiasm of financial institutions to invest in enterprises, and new energy enterprises eventually choose to carry out a green technology innovation strategy.

The change in government strategy choice over time is shown in Figure 12c. When the government’s penalty enforcement intensity factor for financial institutions is low (0–0.4), the financial institutions risk not investing in corporate green technology innovation, the companies choose not to engage in green technology innovation and the government chooses not to engage in environmental regulation due to high cost expenditures and perceived losses. When the penalty intensity factor of the government to financial institutions increases (0.6–1.0), the government evolves towards active environmental regulation. Additionally, the higher the penalty intensity factor, the faster the rate of evolution which eventually reaches stability. Therefore, the government’s green subsidy enforcement intensity factor for financial institutions should be kept at a high level in order to promote a green technological innovation by the enterprises.

The change in financial institution strategy choice over time is shown in Figure 12d. The financial institutions’ strategy choice will generate different behavioral choices as the penalty enforcement intensity changes. When the government penalty enforcement intensity factor for financial institutions is low (0–0.4), the financial institutions are found to bear less losses for not investing in corporate green technology innovation, and therefore financial institutions take the risk of not making green investments. When the penalty enforcement intensity factor is high (0.6–1.0), the financial institutions evolve in the direction of providing green investments to enterprises, and the rate of evolution becomes faster as the penalty enforcement intensity factor increases and eventually reaches a steady state. Therefore, the government imposing high-intensity penalties on financial institutions can better incentivize financial institutions to provide green investment for new energy enterprises.

#### 4.2.3. Impact of Different Intensity Imposition of Carbon Taxation Mechanisms on Evolutionary Paths

The simulation results of the evolutionary game under the variation of the intensity factor of imposition of carbon tax are shown in Figure 13.

The tripartite evolutionary trajectory of the enterprise innovation ecosystem under different intensities of imposition of carbon tax is shown in (a). The government imposes the carbon tax on enterprises, and if enterprises do not engage in green technology innovation, they face the risk of being imposed the tax, which increases their willingness to engage in green technology innovation. When the implementation intensity factor of imposition of carbon tax is small, the enterprises do not choose to carry out green technology innovation and the financial institutions do not invest in green innovation. When the intensity factor of imposition of carbon tax is larger (above 0.6), the enterprises will evolve towards green technology innovation, and the financial institutions will evolve towards investing in enterprises’ green technology innovation, and finally reach stability. That is to say, middle-intensity and high-intensity impositions of carbon tax are able to stimulate enterprises to carry out green technological innovation.

The change of strategy choice of new energy enterprises over time is shown in Figure 13b. When the imposition of carbon tax enforcement intensity factor is small, the enterprises’ willingness to carry out green innovation is enhanced in the short term, but in the long term the enterprises eventually do not choose to carry out green technology innovation; an appropriate increase in the intensity of carbon tax enforcement can facilitate the system’s evolution toward the firms’ choice of green technology innovation strategies, and there is a “threshold” for this facilitation effect. When the intensity of the carbon tax is too low, it is not enough to promote the system to evolve in this direction, but as the intensity of the carbon tax increases, it will drive the system to evolve towards the green technology innovation model. When the enforcement intensity factor reaches 0.6 and above, the enterprises finally stabilize in green technology innovation. Thus, it seems that the imposition of carbon tax can effectively promote enterprises to carry out green technology innovation, and the direct incentive effect is significant. However, this dynamic effect has a tendency to diminish the marginal effect, and the stronger the carbon tax is imposed (over 0.6), the more it is not conducive to the rapid stabilization of enterprises in green technology innovation strategies. Therefore, the intensity of carbon tax should be kept above this threshold (0.6), for too low will fail to promote the effect, but too high will cause a tax burden on enterprises.

The change in government strategy choice over time is shown in Figure 13c. When the levy of carbon tax is relatively low, the perceived loss brought to the government by enterprises not carrying out green technological innovation is far larger than the environmental tax levied on them, and it is impossible to achieve fiscal balance, so the government chooses not to carry out environmental regulation; when the levy of carbon tax is strong (0.6 and above), the government evolves towards positive regulation, and the rate of evolution accelerates with the increase in the levy of environmental tax coefficient and finally reaches stability.

The change of financial institutions’ strategy choice over time is shown in Figure 13d. The financial institutions’ strategic choices result in different behavioral choices in response to changes in the implementation of a carbon tax. When the imposition of a carbon tax is low (0.4 and below), the financial institutions do not invest in corporate green innovation. When the imposition of a carbon tax is high (0.6 and above), the financial institutions evolve towards investing in corporate green innovation. This drive also has a diminishing marginal effect, influenced by corporate strategy. Therefore, the government’s middle-intensity imposition of a carbon tax can well motivate financial institutions to invest in corporate green innovation.

In summary, a carbon tax that is too low may promote the evolution of the system toward collaborative green technology innovation and a carbon tax that is too high may impose a serious tax burden on the enterprise. Therefore, a carbon tax of medium-intensity is most appropriate.

### 4.3. The Effect of Changes in Informal Environmental Regulation Enforcement Intensity Factors on the Evolutionary Path

The simulation results of the evolutionary game under the variation of the intensity factor of informal environmental regulation enforcement are shown in Figure 14.

The tripartite evolutionary trajectory of the enterprise innovation ecosystem under different intensities of informal environmental regulations is shown in Figure 14a. When the intensity factor of the informal environmental regulation is large, the enterprises do not choose to carry out green technology innovation, the financial institutions do not invest in green innovation and the government does not actively carry out environmental regulation. When the intensity factor of the informal environmental regulation is small (0.4 and below), the enterprises evolve toward green technology innovation, the financial institutions evolve toward investing in green technology innovation and the government evolves toward active environmental regulation, eventually reaching stability. In other words, low-intensity informal environmental regulation instruments are able to motivate enterprises to engage in green technology innovation.

The change of strategy choice of new energy enterprises over time is shown in Figure 14b. When the informal environmental regulation implementation intensity factor is small (0–0.4), the enterprises choose to engage in green technology innovation; while when the informal environmental regulation implementation intensity factor is large (0.6–1.0), the enterprises do not choose to engage in green technology innovation. This may be determined by the particularity of the informal environmental regulation tools. As a spontaneous environmental regulation tool, it may promote enterprises to carry out green technology innovation in the early stages of implementation. However, as the cost increases and time passes, the companies’ spontaneous awareness decreases and they do not promote green technology innovation. Thus, it seems that the informal environmental regulation tool does not play a significant role in promoting green technology innovation.

The change in government strategy choice over time is shown in Figure 14c. When the implementation factor of the informal environmental regulation is small (0–0.4), the government needs to bear less costs, so the government chooses to actively carry out environmental regulation; however, because the difference between the benefit and input ratio of informal environmental regulation tools is too large, as the implementation factor of the informal environmental regulation tools increases, the government will gradually reduce its willingness to carry out environmental regulation and eventually stabilize.

The change of financial institutions’ strategy choice over time is shown in Figure 14d. When the informal environmental regulation enforcement intensity factor is small (0–0.4), the enterprises choose to engage in green technology innovation and the financial institutions provide green investment to firms because the perceived benefits are higher than their investment risks; when the informal environmental regulation enforcement intensity factor is large (0.6–1.0), the enterprises do not choose to engage in green technology innovation. The financial institutions’ perceived losses are too high, and considering the investment risks, the financial institutions choose not to provide green investments to the enterprises.

### 4.4. Summary

In summary, both formal and informal environmental regulations can play a role in promoting green technology innovation in enterprises, provided that the implementation intensity is at an appropriate level, as shown in Figure 15. This is also the focus and difficulty of this study. Combined with the conclusions shown in Figure 9, Figure 10, Figure 11, Figure 12, Figure 13 and Figure 14, it is found that the medium-intensity subsidy mechanism (medium-intensity financial support, medium-intensity financial subsidies), the high-intensity penalty mechanism and the medium-intensity carbon tax mechanism under formal environmental regulation represent the optimal execution intensity for encouraging enterprises to carry out green technology innovation; a low-intensity public supervision mechanism under informal environmental regulations is the optimal implementation of green technology innovation. Comparing Figure 9, Figure 10, Figure 11, Figure 12, Figure 13 and Figure 14, in general, formal environmental regulation plays a significantly stronger role than informal environmental regulation, and six types of environmental regulation strategies, namely, “penalty enterprises mechanism“, “financial support mechanism“, “public supervision mechanism”, “punishes financial institutions mechanism”, “financial subsidy mechanism” and “carbon tax mechanism“, have a decreasing effect on promoting the development of the green technology innovation ecosystem of enterprises. Therefore, when promoting the development of a green technology innovation ecosystem of enterprises, the government should first start with policy subsidies to provide a favorable environment for green innovation and green investment for enterprises and green financial institutions, while it is also very necessary to implement regulatory and punitive means to severely crack down on the non-green speculative behaviors of relevant participants, as well as a reasonable carbon tax on businesses.

## 5. Conclusions

Based on the premise that all game subjects are finite rational, this study constructs a game model for the evolution of behavioral strategies of three participating subjects in the green innovation ecosystem of enterprises based on the green technology innovation problems of new energy enterprises and the investment dilemma of financial institutions. We systematically analyze the impact of the different behavioral strategy choices of all parties in the green technology innovation ecosystem of enterprises under dual environmental regulations on green technology innovation of new energy enterprises. Furthermore, we simulate and analyze the influencing factors and evolutionary paths of strategy choices of government, enterprises and financial institutions with a numerical simulation. The study shows that:(1)Both formal and informal environmental regulation by governments can provide some incentives for firms to innovate green technologies. However, the effects of the two types of environmental regulation tools on the green technology innovation ecosystem of enterprises are not the same. The subsidy mechanism under formal environmental regulation can effectively and positively motivate enterprises to carry out green technology innovation and financial institutions to invest in green innovation, thus forming a positive interaction among all participating actors in the enterprise green technology innovation ecosystem. The carbon tax mechanism and punishment mechanism under the formal environmental regulation can produce a pushback effect, prompting enterprises to have sufficient motivation to carry out green technological innovation and financial institutions to have sufficient willingness to invest in the green technological innovation of enterprises.(2)The informal environmental regulation is weaker than the formal environmental regulation in promoting green technological innovation in enterprises, which may be due to an increased public awareness of environmental protection, but is limited by the imperfect feedback mechanism, making it difficult for informal environmental regulation to function effectively.(3)The six types of environmental regulation strategies with different implementation intensity factors have different impacts on the firms’ green technology innovation. The middle-intensity financial support mechanism and the middle-intensity financial subsidy mechanism can better motivate enterprises to carry out green technology innovation and financial institutions to provide green investment for enterprises green technology innovation. The “innovation compensation” effect produced by a low subsidy mechanism is weak and cannot offset the input cost of enterprises; an excessive implementation of the subsidy mechanism is likely to cause excessive government spending and make it difficult to maintain fiscal balance, so a medium-intensity subsidy mechanism can encourage enterprises to stabilize in green technology innovation more rapidly.(4)The high-intensity “penalty enterprises mechanism” and the high-intensity “penalty financial institution mechanism” can force enterprises to invest in green technology innovation and financial institutions to invest in green innovation. The enforcement of the punishment mechanism is too low, resulting in an insufficient crackdown on the non-green speculative behavior of enterprises and financial institutions. Enterprises and financial institutions may seek high profits with low losses, thereby increasing the probability of “cheating“ the government’s green subsidies. The high-intensity penalty mechanism may lead to high penalties for companies and financial institutions for not implementing green innovation, which greatly reduces the probability of companies taking the risks not to innovate green technology and financial institutions taking the risks not to invest in green technology.(5)Implementing a carbon tax and keeping it within a reasonable range can help firms implement green technology innovation. An appropriate increase in the carbon tax rate can facilitate the evolution of the system toward the firms’ choice of green technology innovation strategies, but there is a “threshold” value for this facilitation. When the intensity of the carbon tax is too low, it is not enough to promote the system to evolve in this direction, but as the intensity of the carbon tax increases, it generates a certain driving force for the system to evolve towards the green technology innovation model, and this driving force has a tendency to diminish the marginal effect. Therefore, the carbon tax intensity should be kept above the threshold, as being too low may result in not being able to play a role in promoting the system, and being too high may cause a tax burden on the enterprises. Therefore, middle-intensity carbon tax mechanisms can promote enterprises to stabilize their green technology innovation strategies more rapidly.(6)Low-intensity informal environmental regulations can drive firms to stabilize their technological innovation strategies more rapidly. This may be determined by the particularity of informal environmental regulation tools. As a spontaneous environmental regulation tool, it may promote enterprises to carry out green technology innovation in the early stages of implementation. However, as the cost increases and time passes, the companies’ spontaneous awareness decreases and they do not promote green technology innovation. Thus, low-intensity informal environmental regulations are more likely to promote the green technological innovation by firms.(7)In promoting the development of corporate green technology innovation ecosystems, the “penalty enterprises mechanism “, “financial support mechanism“, “public supervision mechanism”, “punishes financial institutions mechanism”, “financial subsidy mechanism” and “carbon tax mechanism“ are sequentially less effective.(8)The government can more effectively and rapidly promote the green technology innovation of new energy enterprises by choosing the appropriate environmental regulation combination strategy and the appropriate implementation intensity. Combining the implementation of a medium-intensity subsidy mechanism, high-intensity penalty mechanism, a low-intensity public supervision mechanism and a middle-intensity carbon tax mechanism is the optimal strategy combination to encourage enterprises to innovate in green technology, which can motivate enterprises to stabilize in green technology innovation strategies in the shortest time.

## 6. Recommendations

Based on the above research, the following recommendations are made:

First, select the appropriate type of environmental regulation and formulate related policies. Formal environmental regulation can achieve more obvious implementation results, but it also increases the financial burden of enterprises and governments. Informal environmental regulations are cheaper to implement, but less effective than formal environmental regulations. Therefore, we should focus on the synergistic development of formal and informal environmental regulations. While actively implementing formal environmental regulations such as subsidies, penalties and environmental taxes, we should also strengthen the residents’ awareness of environmental protection, increase public participation in environmental protection and improve the feedback mechanism.

Second, strengthen the leading position of the government in the green technology innovation of enterprises and use the double incentive effect to promote the enterprises’ green technology innovation. Reduce the cost of the enterprises’ green technology innovation and the investment cost of the financial institutions through fiscal and tax subsidies, which positively encourages enterprises to carry out green technology innovation; supplemented by appropriate punishment measures, this reversely encourages enterprises and financial institutions to participate in green innovation.

Third, levy a carbon tax in the right range, and implement a system of earmarking carbon tax. Truly tax polluters and cut taxes on green innovative enterprises. In order to avoid the excessive financial burden of a carbon tax levy on enterprises and affect their production process, the government should lower the structural taxes such as income tax and value-added tax for green technology innovation enterprises while levying carbon tax on them to stimulate their innovation vitality.

Fourth, determine the appropriate intensity of environmental regulation. First, low government subsidies are not enough to motivate enterprises and financial institutions to participate in green innovation; too high subsidies will increase the government’s burden and make it difficult for the government to reach an evolutionary stable state. Therefore, only within the appropriate range can we effectively mobilize enterprises to carry out green technology innovation and increase the willingness of financial institutions to invest in green technology innovation. Second, the penalty intensity factor must reach a certain intensity in order to inverse the incentive for enterprises and financial institutions to participate in green technology innovation; otherwise, it will increase the possibility of enterprises and financial institutions accepting subsidies but not fulfilling their responsibilities. Third, the cost of publicizing green ideas should be controlled within a certain range; otherwise, the social benefits generated are far less than the input costs, which is not conducive to the healthy development of the enterprise green innovation ecosystem.

Fifth, foster an active and healthy corporate green innovation ecosystem. Use macro-regulation tools to establish a sound market incentive mechanism and implement the optimal strategy combination to reduce the enterprises’ concerns about the depressed green consumer market and the financial institutions’ concerns about green investment risks. This will maximize the willingness of enterprises to engage in green technology innovation and the willingness of green financial institutions to make green investments to provide a good guarantee for the building of a proven green innovation ecosystem.

From the perspective of government regulation, this paper uses the tripartite evolutionary game theory to discuss how to build an effective enterprise green technology innovation ecosystem; it obtains some important conclusions, but there still are shortcomings. For example, due to the limited conditions for actual research, the simulation parameters are set based on expert opinions rather than actual parameters. Therefore, the simulation graph can only reflect the general trend of the strategic choices of all parties in the green technology innovation ecosystem of enterprises. This is where further in-depth research is needed in the future.

## Figures and Tables

**Figure 1 ijerph-19-11047-f001:**
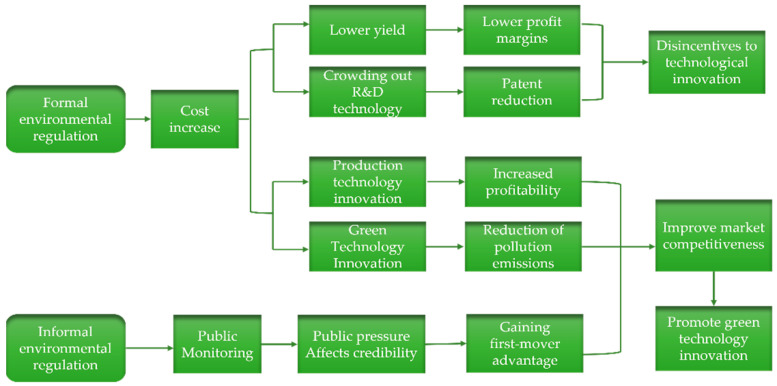
The mechanism of dual environmental regulation on green technology innovation.

**Figure 2 ijerph-19-11047-f002:**
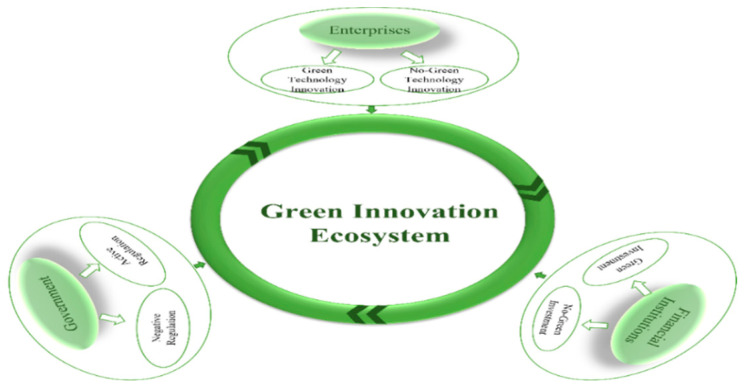
Enterprise green innovation ecosystem.

**Figure 3 ijerph-19-11047-f003:**
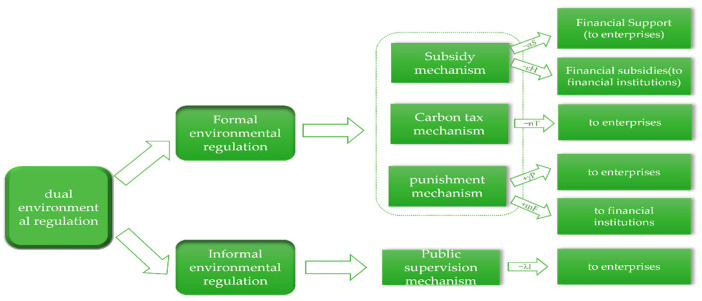
Specific settings for dual environmental regulations.

**Figure 4 ijerph-19-11047-f004:**
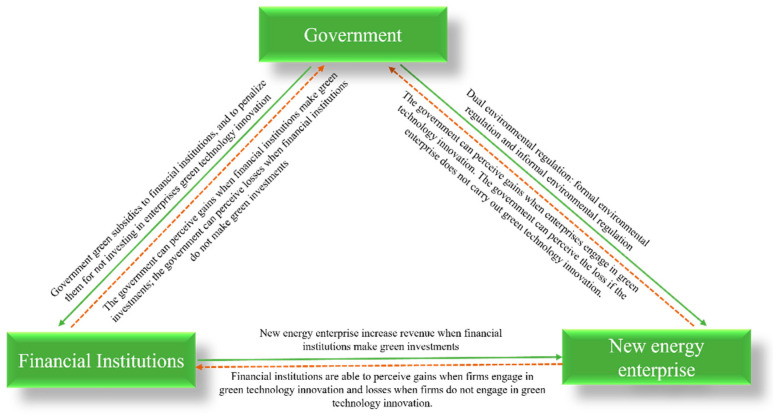
Enterprise green innovation ecosystem impact mechanism.

**Figure 5 ijerph-19-11047-f005:**
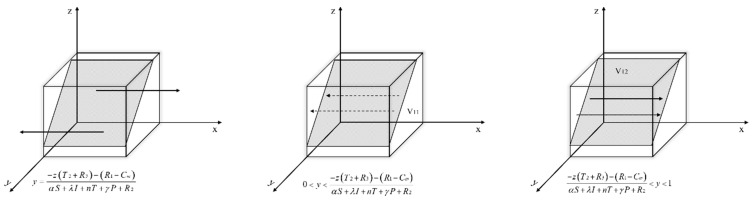
Dynamic evolution of new energy enterprise decision-making.

**Figure 6 ijerph-19-11047-f006:**
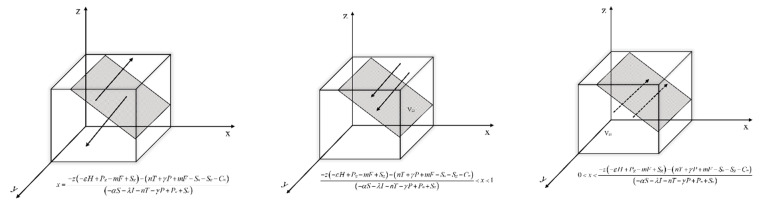
Dynamic evolution of government decision-making.

**Figure 7 ijerph-19-11047-f007:**
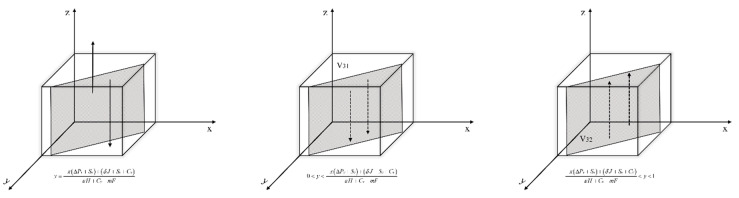
Dynamic evolution of financial institutions decision-making.

**Figure 8 ijerph-19-11047-f008:**
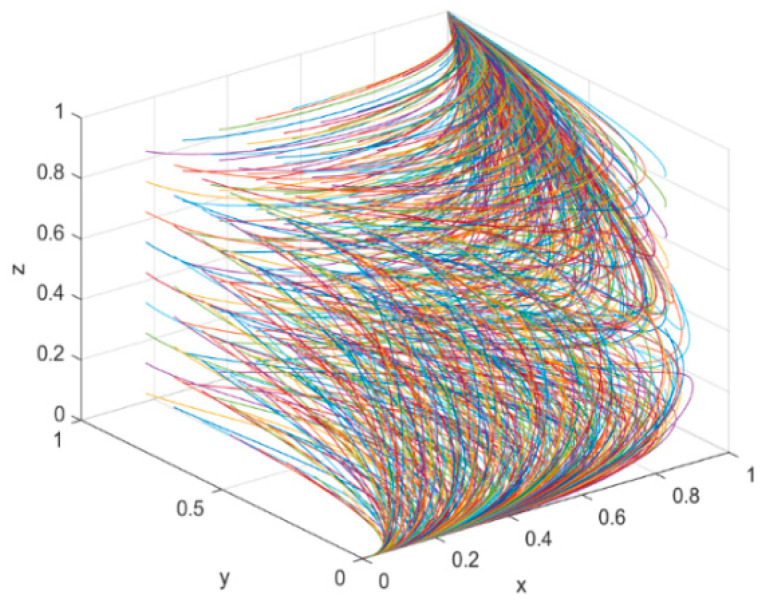
New energy enterprises, government and financial institutions.

**Figure 9 ijerph-19-11047-f009:**
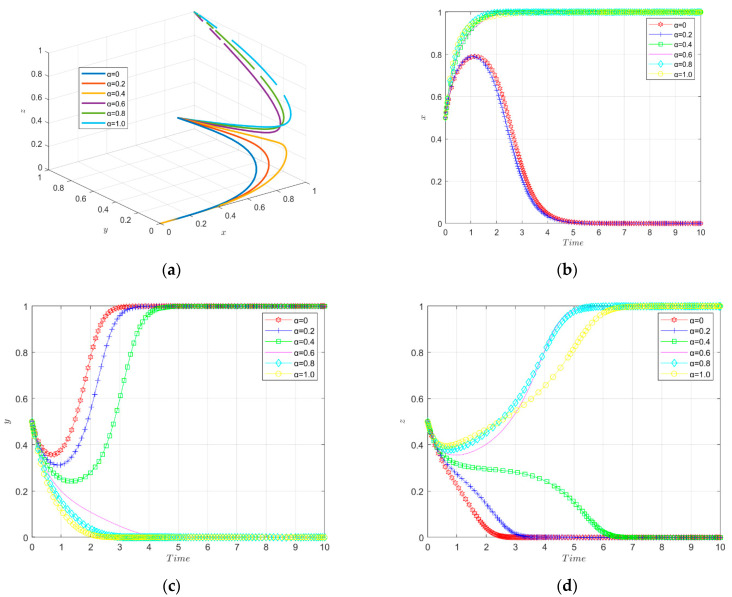
(**a**) Triadic evolutionary trajectory under α changes. Evolutionary trajectories of enterprises (**b**), government (**c**) and financial institutions (**d**) under different financial support enforcement intensities.

**Figure 10 ijerph-19-11047-f010:**
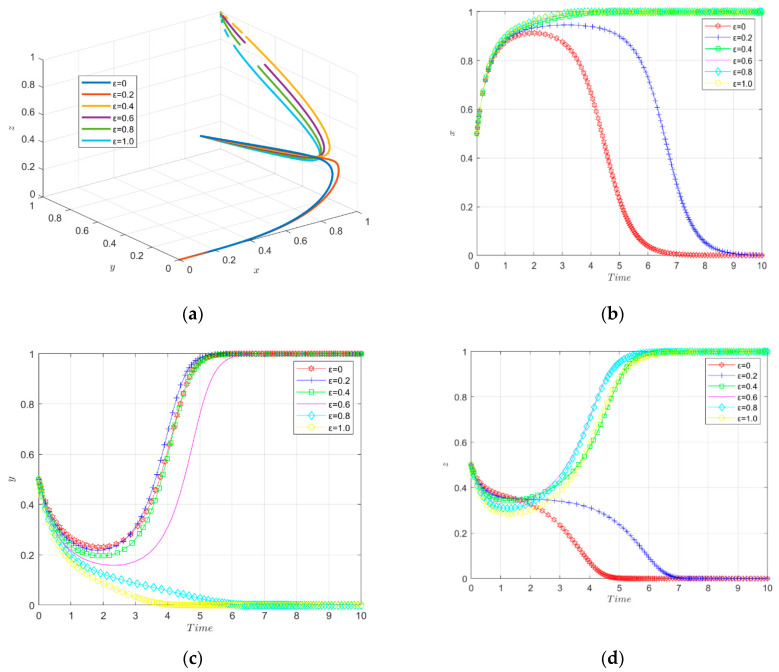
(**a**) Evolutionary trajectory of the tripartite game under different financial subsidies intensities. Evolutionary trajectories of enterprises (**b**), government (**c**) and financial institutions (**d**) under different financial subsidies intensities.

**Figure 11 ijerph-19-11047-f011:**
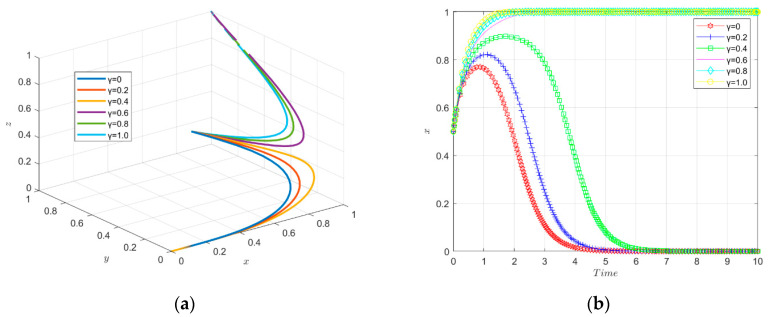
(**a**) Triadic evolutionary trajectory under γ changes. Evolutionary trajectories of enterprises (**b**), government (**c**) and financial institutions (**d**) under different penalties on enterprises enforcement intensities.

**Figure 12 ijerph-19-11047-f012:**
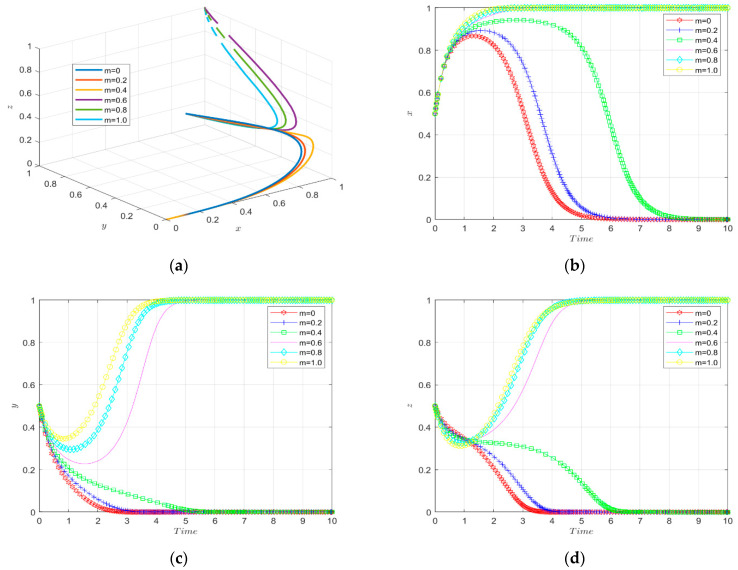
(**a**) Evolutionary trajectory of the tripartite game under different penalty intensities. Evolutionary trajectories of enterprises (**b**), government (**c**) and financial institutions (**d**) under different penalty intensities.

**Figure 13 ijerph-19-11047-f013:**
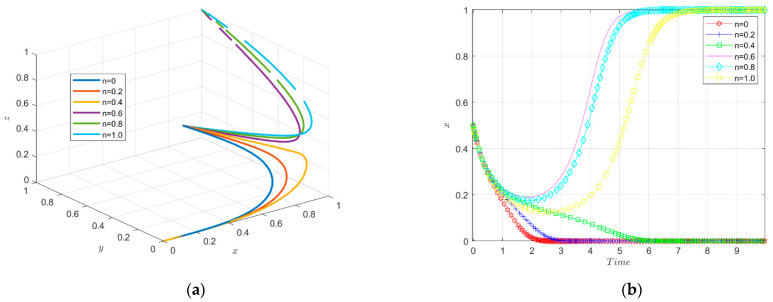
(**a**) Evolutionary trajectory of the tripartite game under n changes. Evolutionary trajectories of enterprises (**b**), government (**c**) and financial institutions (**d**) under different carbon taxes.

**Figure 14 ijerph-19-11047-f014:**
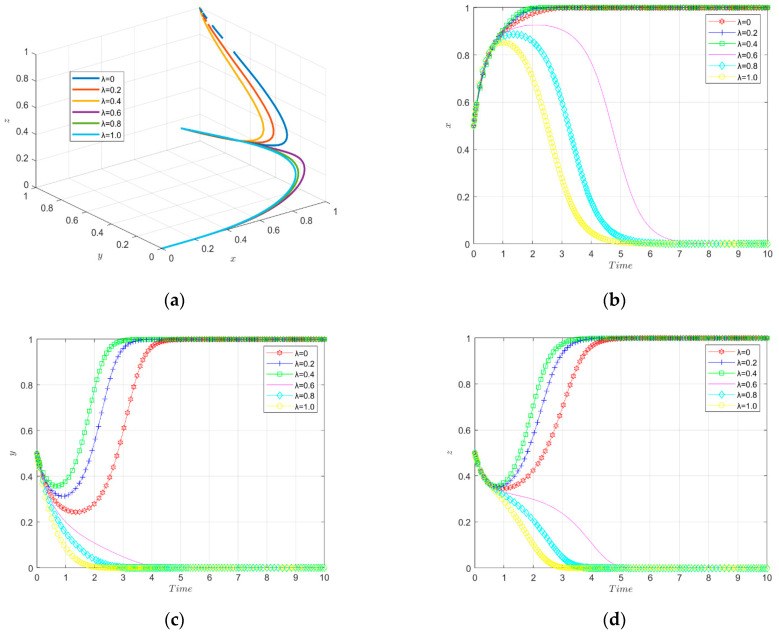
(**a**)Triadic evolutionary trajectory under λ changes. Evolutionary trajectories of enterprise (**b**), government (**c**) and financial institutions (**d**) under different informal environmental regulation enforcement intensities.

**Figure 15 ijerph-19-11047-f015:**
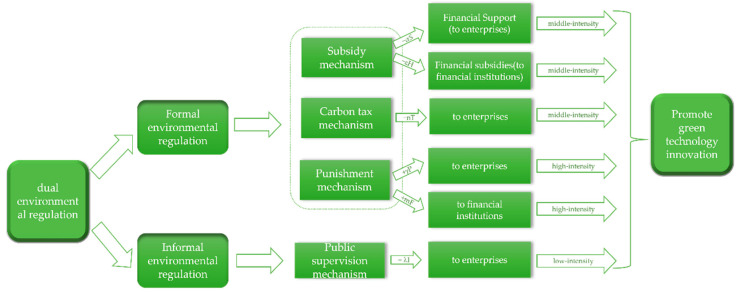
Dual environmental regulation that can promote the optimal implementation of green technology innovation.

**Table 1 ijerph-19-11047-t001:** Parameter symbols and meanings.

Symbols	Measure
S	Financial support ceiling
α	Financial support execution intensity factor
T	Cap on carbon taxation
n	Carbon tax implementation intensity factor
P	The government’s penalty cap on new energy companies receiving financial support without green technology innovation behavior
γ	Government penalty intensity factor for new energy enterprises
I	Informal environmental regulation expenditure cap
λ	The government communicates the implementation intensity factor to the public
J	Financial institutions provide green investment caps to new energy enterprises
δ	Green investment intensity factor of financial institutions in enterprises
H	Government financial subsidies to financial institutions cap
ε	Government financial subsidies intensity factor for financial institutions
F	The government’s penalty cap on financial institutions receiving green subsidies but not investing in corporate green innovation behavior
m	Government penalty intensity factor for financial institutions
R	Basic earnings of new energy enterprises
R_1_	Increased revenue when enterprises innovate green technologies
R_2_	New energy enterprises increased revenue when government actively regulates
R_3_	New energy enterprises increased revenue when financial institutions green invest
C_m_	Input costs for green technology innovation in new energy enterprises
C_n_	Government input costs when choosing an active regulation strategy
C_y_	Costs invested by financial institutions in monitoring enterprises
T_2_	Losses borne by enterprises whose reputation is damaged when they receive green investments from financial institutions but do not carry out green technology innovation
P_n_	The perceived benefits to government when enterprises engage in green technology innovation
S_n_	Perceived loss to government when enterprises do not carry out green technology innovation
P_y_	Basic earnings when financial institutions do not green invest
ΔP_y_	Perceived benefits to financial institutions when companies engage in green technology innovation
S_y_	Perceived loss to financial institutions when enterprises do not carry out green technology innovation
P_g_	Perceived benefits to government when financial institutions choose to invest
S_g_	Government perceived loss when financial institutions do not invest in green
x, y, z	Tri-party behavioral strategy options for enterprises, government and financial institutions

**Table 2 ijerph-19-11047-t002:** Payoff matrix among the new energy enterprises, government and financial institutions.

Financial Institutions	New Energy Enterprises	Government					
		Active regulation (y)			Negative regulation (1 − y)		
		Government Payoff	Enterprises Payoff	Financial Institutions payoff	Government Payoff	Enterprises Payoff	Financial Institutions Payoff
Green Investment (z)	Green technology innovation (x)	P_g_ + P_n_ − αS − λI − εH − C_n_	αS + λI +δJ + R + R_1_ + R_2_ + R_3_ − C_m_	P_y_ + ΔP_y_ − δJ + εH	0	δJ + R + R_1_ + R_3_ − C_m_	P_y_ + ΔP_y_ − δJ − C_y_
	No green technology innovation (1 − x)	nT + γP + P_g_ − εH − S_n_ − C_n_	R + δJ −nT −γP − T_2_	P_y_ + εH − δJ − S_y_	0	δJ + R − T_2_	P_y_ − δJ − C_y_ − S_y_
No green Investment (1 − z)	Green technology innovation (x)	mF + P_n_ − αS − λI − S_g_ − C_n_	αS + λI + R + R_1_ + R_2_ − C_m_	P_y_ − mF	0	R + R_1_ − C_m_	P_y_
	No green technology innovation (1 − x)	nT + γP + mF − S_n_ − S_g_ − C_n_	R − nT − γP	P_y_ − mF	0	R	P_y_

**Table 3 ijerph-19-11047-t003:** Eigenvalues of Jacobi matrix.

Equilibrium Points	λ_1_	λ_2_	λ_3_
E_1_ = (0, 0, 0)	R_1_ − C_m_	nT + γP + mF − S_n_ − S_g_ − C_n_	−(δJ + S_y_ + C_y_)
E_2_ = (0, 1, 0)	αS + λI + nT + γP + R_2_ + R_1_ − C_m_	S_n_+ S_g_ + C_n_ − nT − γP − mF	εεH − mF − δJ − S_y_
E_3_ = (0, 0, 1)	T_2_ + R_1_ + R_3_ − C_m_	nT + γP + P_g_ − εH − S_n_ − C_n_	δJ + S_y_ + C_y_
E_4_ = (1, 0, 0)	−(R_1_ − C_m_)	mF + P_n_ − αS − λI − S_g_ − C_n_	ΔP_y_ − δJ − C_y_
E_5_ = (0, 1, 1)	T_2_ + αS +λI + nT + γP + R_1_ + R_2_ + R_3_−C_m_	εH + S_n_ + C_n_ − nT − γP − P_g_	mF + δJ + S_y_ − εH
E_6_ = (1, 1, 0)	C_m_ − αS − λI − nT− γP − R_2_ − R_1_	αS + λI + S_g_ + C_n_ − mF − P_n_	ΔP_y_ + εH − mF − δJ
E_7_ = (1, 0, 1)	C_m_ − T_2_ − R_1_ − R_3_	−εH − αS − λI + P_g_ + P_n_ − C_n_	δJ + C_y_ − ΔP_y_
E_8_ = (1, 1, 1)	−(T_2_ + αS +λI + nT + γP + R_1_ + R_2_ + R_3_ − C_m_)	−(−εH − αS − λI + P_g_ + P_n_ − C_n_)	−(ΔP_y_ + εH − mF − δJ)
E_9_ = (x*, y*, z*)	Saddle point		

**Table 4 ijerph-19-11047-t004:** Stability of the replicated power system.

	**Situation I:** **R_1_ − C_m_ > 0** **nT + γP + mF−S_n_ − S_g_ − C_n_ > 0**				**Situation II:** **R_1_ − C_m_ > 0** **nT + γP + mF−S_n_ − S_g_ − C_n_ < 0**			
**Equilibrium Points**	**λ_1_**	**λ_2_**	**λ_3_**	**State**	**λ_1_**	**λ_2_**	**λ_3_**	**State**
E_1_ = (0, 0, 0)	+	+	−	Instability point	+	−	−	Instability point
E_2_ = (0, 1, 0)	+	−	±	Saddle point	+	+	±	Saddle point
E_3_ = (0, 0, 1)	+	±	+	Saddle point	+	±	+	Saddle point
E_4_ = (1, 0, 0)	−	±	+	Saddle point	−	±	+	Saddle point
E_5_ = (0, 1, 1)	+	±	±	Saddle point	+	±	±	Saddle point
E_6_ = (1, 1, 0)	−	±	+	Saddle point	−	±	+	Saddle point
E_7_ = (1, 0, 1)	−	+	−	Instability point	−	+	−	Instability point
E_8_ = (1, 1, 1)	−	−	−	**ESS**	−	−	−	**ESS**
E_9_ = (x*, y*, z*)	DetJ < 0 ∩ TrJ = 0			Saddle point	DetJ < 0 ∩ TrJ = 0			Saddle point
	**Situation III:** **R_1_ − C_m_ < 0** **nT + γP + mF− S_n_ − S_g_ − C_n_ > 0**				**Situation IV:** **R_1_ − C_m_ < 0** **nT + γP + mF− S_n_ − S_g_ − C_n_ < 0**			
**Equilibrium Points**	**λ_1_**	**λ_2_**	**λ_3_**	**State**	**λ_1_**	**λ_2_**	**λ_3_**	**State**
E_1_ = (0, 0, 0)	−	+	−	Instability point	−	−	−	**ESS**
E_2_ = (0, 1, 0)	+	−	±	Saddle point	+	+	±	Saddle point
E_3_ = (0, 0, 1)	±	±	+	Saddle point	±	±	+	Saddle point
E_4_ = (1, 0, 0)	+	±	+	Saddle point	+	±	+	Saddle point
E_5_ = (0, 1, 1)	+	±	±	Saddle point	+	±	±	Saddle point
E_6_ = (1, 1, 0)	−	±	+	Saddle point	−	±	+	Saddle point
E_7_ = (1, 0, 1)	±	+	−	Saddle point	±	+	−	Saddle point
E_8_ = (1, 1, 1)	−	−	−	**ESS**	−	−	−	**ESS**
E_9_ = (x*, y*, z*)	DetJ < 0 ∩ TrJ = 0			Saddle point	DetJ < 0 ∩ TrJ = 0			Saddle point

## Data Availability

Not applicable.

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
