# Peer review of "Dynamic Game Analysis of Enterprise Green Technology Innovation Ecosystem under Double Environmental Regulation"

_ijerph, 2022, doi:10.3390/ijerph191711047_

Round 1
Reviewer 1 Report
The authors construct a tripartite evolutionary game model of enterprises, governments and financial institutions, and use evolutionary game theory and MATLAB simulation to analyze the evolutionary process of the interaction of the subjects of green technology innovation of enterprises under the dual environmental regulation. I suggest moving all the calculations to a specific mathematical appendix. Also to describe more in details the characteristics of the model and Matlab simulation.
Reviewer 2 Report
Please discuss the differences between your paper and the paper:An Evolutionary Game Analysis on Green Technological Innovation of New Energy Enterprises under the Heterogeneous Environmental Regulation Perspective, Sustainability 2022, 14(10), 6340 because there is a high rate of similarity between this paper and the paper mentioned above.
Reduce the similarity rate. Cite the article mentioned and explain the difference between them. There's similarity and one-on-one quotation at every stage, including methodology
Round 2
Reviewer 2 Report
Corrections are made following the critiques in the previous round. I am happy with the final version and my recommendation is positive.